# Tracking interspecies transmission and long-term evolution of an ancient retrovirus using the genomes of modern mammals

**William E Diehl[†], Nirali Patel[‡], Kate Halm, Welkin E Johnson***

Biology Department, Boston College, Chestnut Hill, United States

**Abstract** Mammalian genomes typically contain hundreds of thousands of endogenous retroviruses (ERVs), derived from ancient retroviral infections. Using this molecular 'fossil' record, we reconstructed the natural history of a specific retrovirus lineage (ERV-Fc) that disseminated widely between ~33 and ~15 million years ago, corresponding to the Oligocene and early Miocene epochs. Intercontinental viral spread, numerous instances of interspecies transmission and emergence in hosts representing at least 11 mammalian orders, and a significant role for recombination in diversification of this viral lineage were also revealed. By reconstructing the canonical retroviral genes, we identified patterns of adaptation consistent with selection to maintain essential viral protein functions. Our results demonstrate the unique potential of the ERV fossil record for studying the processes of viral spread and emergence as they play out across macro-evolutionary timescales, such that looking back in time may prove insightful for predicting the long-term consequences of newly emerging viral infections.

**\*For correspondence:** welkin.
johnson@bc.edu

**Present address:** [†]Program in
Molecular Medicine, University of
Massachusetts Medical School,
Worcester, United States; [‡]M.D.
Program, New Jersey Medical
School, Rutgers, New Jersey,
United States

**Competing interests:** The
authors declare that no
competing interests exist.

**Reviewing editor:** Stephen P
Goff, Howard Hughes Medical
Institute, Columbia University,
United States

## Introduction

Retroviruses (family *Retroviridae*) are abundant in nature and include human immunodeficiency viruses (HIV-1 and HIV-2), human T-cell leukemia viruses (HTLV-1 and -2), and the well-studied oncogenic retroviruses of mice and other model organisms, among many others (*Goff, 2007*). The hallmark of all retroviruses is reverse transcription of the viral RNA genome to form a DNA provirus, which is inserted at random into host chromosomal DNA. If integration of this viral DNA occurs in the germ line, the resulting insertion is called an endogenous retrovirus (ERV). The inserted sequence (the ERV) is replicated as part of the host genome during cell division and can be inherited vertically in a Mendelian fashion. Each ERV integrant is subject to drift and selection and may be lost or, given enough time, become fixed in the population. Over many millions of years, and through repeated rounds of endogenization and copy number expansion, metazoan genomes have become riddled with the remnants of past retroviral infections; in most organisms (including humans), ERVs amount to hundreds of thousands of copies per genome (*Lindblad-Toh et al., 2005*; *Lander et al., 2001*; *Waterston et al., 2002*). Following endogenization, ERV sequences switch from evolving at the very rapid rate associated with exogenous retroviral replication to a rate approximating the neutral evolutionary rate of the host genome. Thus, ERV sequences embedded in animal genomes serve as long-lasting molecular fossils related to exogenous retroviruses and their ancient, extinct relatives.

The group of related ERV elements collectively referred to as ERV-Fc are distantly related to extant gammaretroviruses (*Jern et al., 2005*) and form a monophyletic clade with the human ERVs, HERV-H and HERV-W. All ERV-Fc elements possess a simple genome consisting of the canonical

**eLife digest** Viruses have been with us for billions of years, and exist everywhere in nature that life is found. Viruses therefore have had a significant impact on the evolution of all organisms, from bacteria to humans. Unfortunately, viruses do not leave fossils, and so we know very little about how viruses originate and evolve over time. Fortunately, over the course of millions of years, genetic sequences from the viruses accumulate in the DNA genomes of living organisms (including humans). These sequences can serve as molecular "fossils" for exploring the natural history of viruses and their hosts.

Diehl et al. have now searched the genomes of 50 modern mammals for "fossil" viral remnants of an ancient group of viruses known as ERV-Fc. This revealed that ERV-Fc viruses infected the ancestors of at least 28 of these mammal species between 15 million and 30 million years ago. The viruses affected a diverse range of hosts, including carnivores, rodents and primates. The distribution of ERV-Fc among different mammals indicates that the viruses spread to every continent except Antarctica and Australia, and that they jumped between species more than 20 times.

Diehl et al. also pinpointed patterns of evolutionary change in the genes of the ERV-Fc viruses that reflect how the viruses adapted to different host mammals. As part of this process, the viruses often exchanged genes with each other and with other types of viruses. Such genetic recombination is likely to have played a significant role in the evolutionary success of the ERV-Fc viruses.

Mammalian genomes contain hundreds of thousands of ancient viral fossils similar to ERV-Fc. Future work could study these to improve our understanding of when and why new viruses emerge and how long-term contact with viruses affects the evolution of their host organisms.

*gag*, *pro*, *pol*, and *env* genes common to all retroviruses but lack additional regulatory or accessory genes associated with complex retroviruses (*Figure 1*). The designation *ERV-Fc* is based on the practice of naming ERV groups after the tRNA complementarity of the viral primer binding sequence (PBS); in the case of ERV-Fc, the PBS is complementary to a phenylalanine tRNA (GAA anticodon). This viral lineage was first identified and characterized from the genomes of several primate species, including humans, chimpanzees, gorillas, baboons, and multiple New World monkeys (*Bénit et al., 2003*). Estimates of insertion timing suggested independent endogenization in the different primate lineages studied rather than cospeciation after colonization of a common ancestor, and the authors hypothesized that ERV-Fc first infected the common ancestor of all simians and remained actively infectious/mobile for tens of millions of years (*Bénit et al., 2003*). A more recent study described abundant representation of ERV-Fc sequences in the canine genome, and the authors suggested that an ancient cross-species transmission between carnivores and primates could account for the presence of ERV-Fc sequences in the two lineages (*Barrio et al., 2011*).

Our goal in the present study was to reconstruct the natural history of a specific exogenous retrovirus lineage, which gave rise to the ERV-Fc group of ERV loci. Because the various mechanisms that influence post-endogenization sequence evolution and copy-number expansion in organismal genomes can erase or alter ERVs in ways that do not accurately reflect the exogenously replicating progenitor virus, we first sought to minimize the effects of post-endogenization evolution. To do this, we first performed an exhaustive search of mammalian genome sequence databases for ERV-Fc loci and then compared the recovered sequences. Next, for each mammalian genome with sufficient ERV-Fc sequence, we reconstructed Gag, Pol, and Env weighted consensus protein sequences representing the exogenous virus that colonized that particular species' ancestors. Finally, we used these consensus sequences to infer the natural history and evolutionary relationships of the exogenous, ERV-Fc related viruses. In so doing, we uncovered a complex evolutionary history, including a prolonged, ancient global spread of the virus involving multiple instances of cross-species transmission and endogenization, and revealed that recombination played a significant part in the evolution and spread of the ERV-Fc lineage.

**Figure 1.** Schematic representation of the major features of ERV-Fc proviruses. The region colored in blue indicates *gag*, brown indicates *pol*, and yellow represents *env* coding regions. The gray-colored regions indicate the two long terminal repeat (LTR) regions. Vertical lines within these regions indicate where proteolytic cleavage would occur between protein subunits. The identity of these subunits is indicated below the schematic: MA = matrix, CA = capsid, NC = nucleocapsid, PR = protease, RT = reverse transcriptase, IN = integrase, SU = surface, TM = transmembrane, PPT = polypurine tract. The probable location of the viral RNA packaging motif is indicated by ψ. At the termini of the retroviral LTR sequences is shown the canonical TG/CA dinucleotides as well as the 5 nucleotide target site duplications (TSDs) flanking the provirus. ERV, endogenous retrovirus.

## Results

### ERV-Fc sequences are widely distributed among mammalian genomes

Using BLASTn and previously reported ERV-Fc sequences as initial queries, we screened the non-redundant (nr) database and 50 mammalian genome sequence databases ranging in completeness from the nearly complete human and mouse genomes to low-coverage genomic scaffolds and unscaffolded trace and contig archives (*Figure 2* and *Figure 2—source data 1* ) (*Bénit et al., 2003*). Preliminary amino acid phylogenies of translated consensus sequences generated from the initial BLAST hits were used to confirm or exclude ERV-Fc evolutionary relationships. To extract maximal ERV-Fc sequence information from the genomic databases, an iterative BLAST approach was then undertaken using preliminary hits as query sequences. This approach resulted in the identification of ERV-Fc coding sequences in 28 species, representing every superorder of eutherian mammals except Xenarthra (*Figure 2*). No evidence was found for ERV-Fc being present in metatherian mammals. In several cases, a genome possessed evidence of ERV-Fc endogenization, but lacked sufficient sequence information for definitive phylogenetic analysis of Gag/CA, Pol/RT, or Env/TM (*Figure 2—source data 2*). These included the genomes of the Chinese hamster and European shrew that harbor *gag* sequence fragments that branch with ERV-Fc, but are too fragmented to reconstruct complete CA ancestral coding sequences. At the time of sampling, the Chinese hamster and European shrew genomes lacked ERV-Fc *pol* or *env* sequences. Similarly, we found that the orangutan genome harbors a single ERV-Fc-associated solo long terminal repeat (LTR) element (*Figure 2* and *Figure 2—source data 2*).

While ERV-Fc was present in the majority of mammalian species examined, its absence from the genomes of several eutherian lineages, such as New World rodents (degu, chinchilla, guinea pig) and ruminants (sheep, cow, water buffalo), is inconsistent with a single endogenization event in a common ancestor of all eutherian mammals. Additionally, the genomes of several species, including multiple primate and carnivore species, contained multiple genetically distinct ERV-Fc lineages (*Figure 2* and *Figure 2—source data 2*). Combined, these findings are consistent with a natural history marked by numerous cross-species transmissions leading to independent episodes of genome colonization in the ancestors of the examined species (see subsequent section on cross-species transmission).

Similar to most ancient ERV loci, the viral open reading frame (ORF) sequences present in the vast majority of retrieved ERV-Fc elements are disrupted by mutations (including point-mutations, insertions, and deletions), precluding expression of functional viral proteins. However, we found intact ORFs corresponding to the viral *env* gene in several species (indicated by an envelope icon in *Figure 2*). Based on sequence inspection, these are ORFs that potentially retain the capacity to encode retroviral envelope glycoproteins. Species with open *env* ORFs include aardvark, gray mouse lemur, squirrel monkey, marmoset, baboon, chimpanzee, human, dog, and panda. With one exception, each of these ORFs is unique to the species in which it was identified, indicating independent origins for each. The exception is the previously characterized HERV-Fc1[env] locus (*Bénit et al., 2003*), which is present in the genomes of all great apes. The *env* ORF of this locus is open in human, chimpanzee, and bonobo orthologs, whereas mutations have disrupted the ORF in the gorilla ortholog.

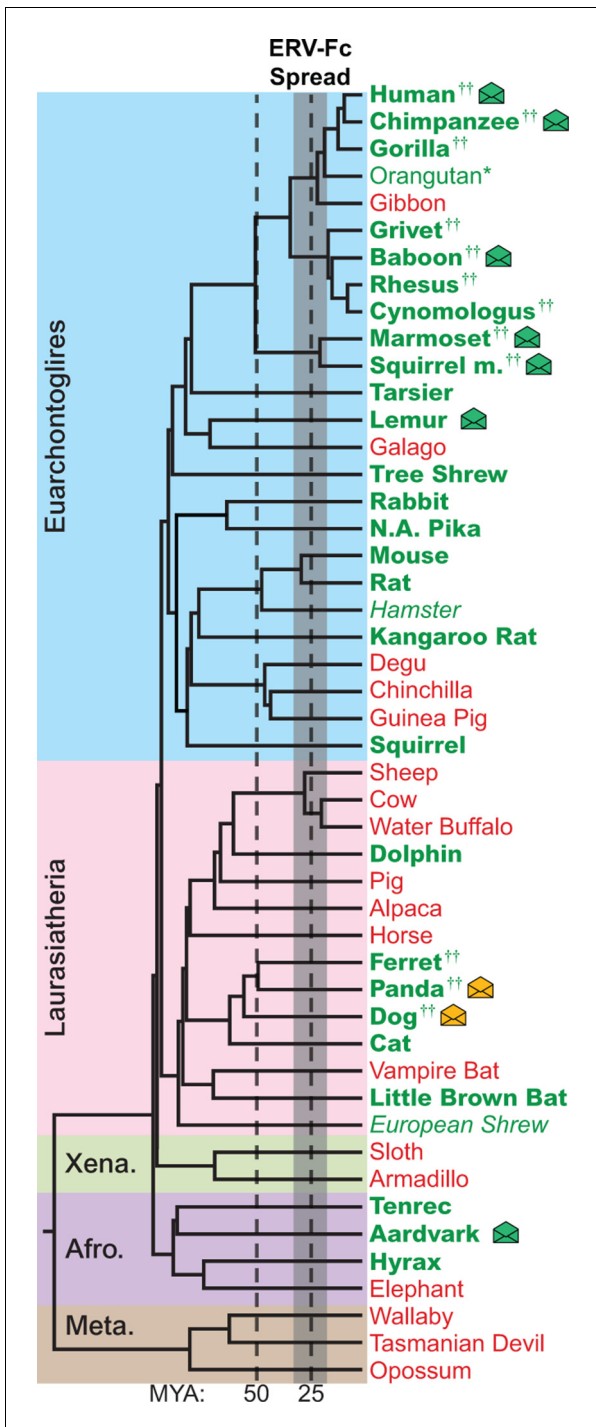

**Figure 2.** The genomes of most Eutherian mammals harbor ERV-Fc. A mammalian phylogeny (adapted from [(**Bininda-Emonds et al., 2007**]) including species whose genomes were examined for the presence of ERV-Fc. Species lacking ERV-Fc are depicted in red, while those found to harbor ERV-Fc are depicted in green. Bold font indicates that coding potential in one or more gene regions could be reconstructed; italics indicates that ERV-Fc fragments were identified but coding potential could not be reconstructed; * indicates that only a solo LTR was identified; and †† indicates that a species harbors two genetically distinct ERV-Fc lineages. Background shading indicates higher-order taxonomic relationships: blue = Euarchontoglires, pink = Laurasiatheria, green = Xenarthra, purple = Afrotheria, brown = Metatheria. Envelope icons indicate species in which ERV-Fc *env* open reading frame (s) were identified, and the icons colored green indicate *env* with homology to HERV-Fc; yellow icons indicate the *env* had greater similarity to HERV-W. ERV, endogenous retrovirus. HERV, human ERV.

*Figure 2 continued on next page*

*Figure 2 continued*

The following source data is available for figure 2:

**Source data 1.** Genome sequence database summary.
**Source data 2.** Overview of recovered ERV-Fc sequences.
**Source data 3.** Sequences of ERV-Fc primer binding sites.

## Reconstruction of viral genomes and proteins reveals evolutionary signatures of significant exogenous ERV-Fc replication

The primary goal of this study was to use ERV-Fc sequences to gain insight into the nature of the related exogenous viral agents that infected and spread among mammalian hosts. Indeed, by examining our consensus reconstructions in light of the known functions of each viral protein and their roles in replication, we found a number of patterns most consistent with extensive spread and evolution of an exogenous retrovirus (described in detail in subsequent sections). This indicates that our reconstructions reflect the nature of the exogenous agent that left ERV-Fc sequences behind in the germlines of its mammalian hosts.

### Proviral structure

The majority of identified ERV-Fc loci with intact 5' and 3' ends also possess canonical (5') TG...CA (3') dinucleotides at the termini of both LTRs (*Figure 1*), and post-integration mutations likely account for the loci that do not fit this pattern. Where availability of sequence allowed, the identity of the viral primer-binding site (PBS) was confirmed to be complementary to a GAA anticodon tRNA[Phe] (*Figure 2—source data 3*). Consistent with previously published observations, all ERV-Fc sequences possessed a distinct bias in nucleotide content, with an overrepresentation of cytosine residues indicating that this is a conserved feature for the entire ERV-Fc lineage (*Jern et al., 2005*). Similar to other gammaretroviruses, ERV-Fc genomes contain a single polypurine tract immediately upstream of the 3' LTR (*Figure 1*). In all cases where ERV-Fc-associated LTRs could be identified, 5 base-pair (b.p.) target site duplications (TSDs) of host DNA were found flanking the provirus. This feature of ERV-Fc differs from extant mammalian gammaretroviruses and most gamma-like (Class I) ERVs, which produce 4-b.p. TSDs. Gamma-like retroviruses that are known to generate 5-b.p. TSDs include spleen necrosis virus (SNV), reticuloendotheliosis virus (REV), and the HERV-H group of human ERV [*Ballandras-Colas et al., 2013*; *Derse et al., 2007*; *Holman and Coffin, 2005*; *Kim et al., 2010*; *Shimotohno and Temin, 1980*, W.E. Diehl unpublished data].

Similar to extant gammaretroviruses and gamma-like ERVs, ERV-Fc consensus genomes contain in-frame *gag-pro-pol* sequences (*Figure 3A*), where production of the combined Pro-Pol polyprotein results from termination suppression of the *gag* stop codon (*Swanstrom et al., 1997*). As with other gammaretroviruses, none of the ERV-Fc sequences appear to encode accessory genes. In most cases, reconstructed ERV-Fc genomes possessed an *env* ORF that overlaps with the *pol* ORF, but is encoded in an alternate reading frame. Many extant gammaretroviruses share this genomic architecture, including murine leukemia virus (MLV) (*Shinnick et al., 1981*). We found exceptions to this architecture in Old World primate and hominid ERV-Fc2 sequences, where the encoded *env* genes do not overlap with *pol* (*Figure 3A*).

### ERV-Fc structural proteins

Based on consensus proteins, the predicted ERV-Fc Gag polyprotein sequences resemble that of other gammaretroviruses in that they comprise three viral proteins corresponding to matrix (MA), capsid (CA), and nucleocapsid (NC), as well as a spacer peptide between MA and CA similar to the p12 protein of MLV (*Figure 3A*). Henceforth, we refer to this region as 'p12' based on its positional homology to MLV p12, although the predicted molecular weights of the peptides encoded by ERV-Fc *gag* genes are not 12 kDa.

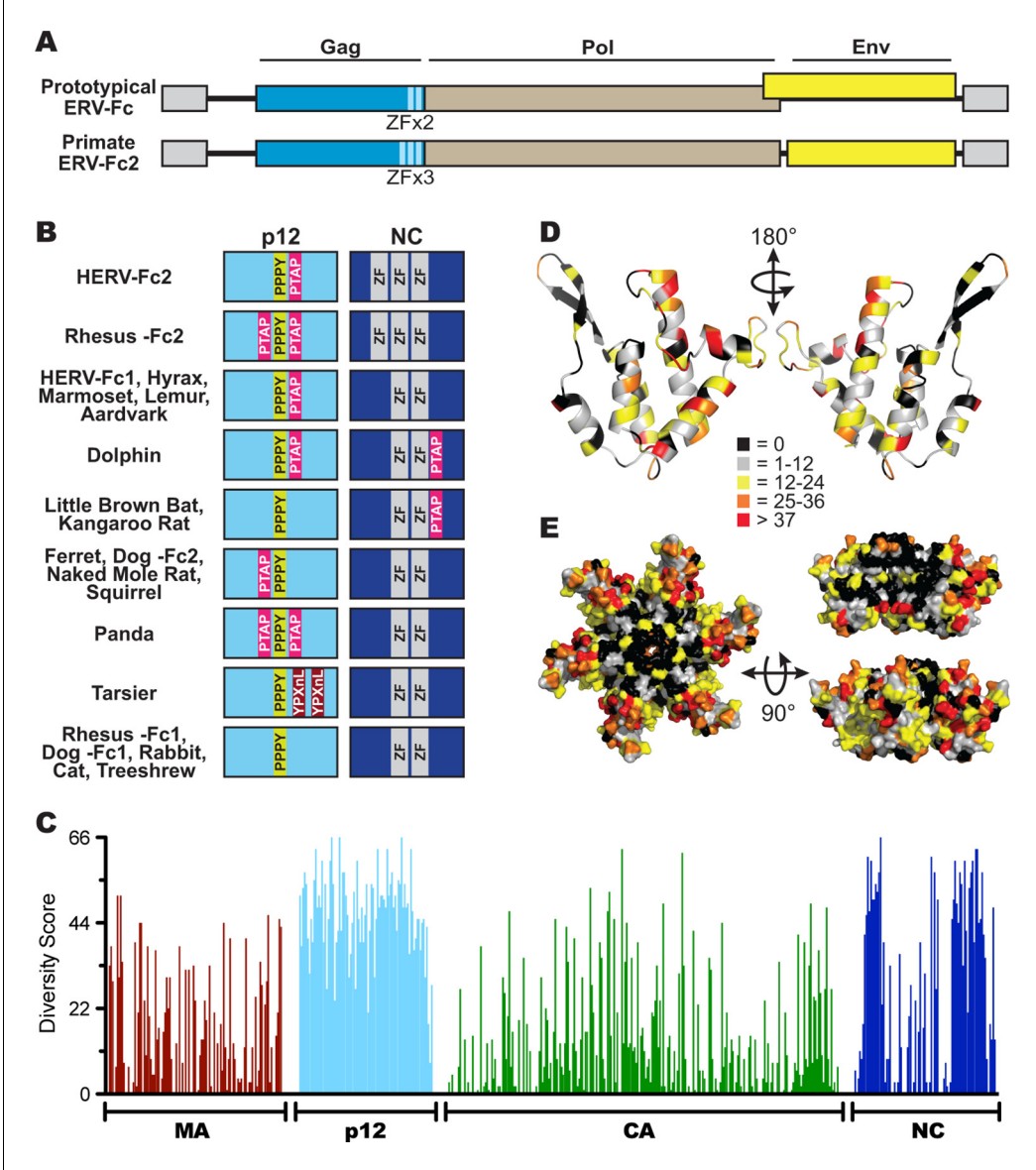

**Figure 3.** Sequence diversity in ERV-Fc is consistent with an extended period of exogenous replication. (**A**) Structures of ERV-Fc genomes with blue, brown, and yellow boxes indicating *gag, pol,* and *env* coding sequences, respectively. Light blue regions indicate the multiple zinc finger motifs in the NC subunit. (**B**) Organization of late domains (PPPY, PTAP, YPXnL) and zinc finger domains within the p12 and NC subunits of ERV-Fc Gag, respectively. (**C**) Plot of amino acid diversity across ERV-Fc Gag with the diversity score calculated by summation of 'match scores' (*Smith and Smith, 1990*) in pairwise comparisons of ERV-Fc sequences to a global consensus sequence. (**D**) Ribbon diagram of ERV-Fc consensus monomeric CA model, with the residues highlighted according to their diversity score. (**E**) Surface view of hexameric ERV-Fc consensus N-terminal CA domain model: left = view of cytoplasmic exposed surface; top right = cross-sectional view through hexamer, with three monomers removed; bottom right = surface view of monomers available for interhexamer interactions. In both right panels, the figure is oriented such that the cytoplasmic exposed surface of CA is at the top.

The following figure supplements are available for figure 3:

**Figure supplement 1.** Amino acid diversity in ERV-Fc Pol.

**Figure supplement 2.** Amino acid diversity in ERV-Fc Env.

The retroviral MA protein plays a critical role in retroviral assembly, mediating association of the viral Gag molecules with the plasma membrane of the host cell. This interaction is mediated by two essential, and highly conserved, structural motifs: a myristoyl moiety added to the N-terminal glycine of the mature protein that becomes embedded in the lipid membrane (*Henderson et al., 1983*; *Ootsuyama et al., 1985*; *Rein et al., 1986*; *Schultz and Oroszlan, 1983*; *Veronese et al., 1988*), and an adjacent polybasic motif that mediates interaction with the charged head groups of the plasmid membrane (*Murray et al., 2005*). All ERV-Fc MA consensus sequences identified in this study possessed an N-terminal glycine that would be predicted to receive a myristoyl modification during protein generation. Similarly, all ERV-Fc MA proteins also contained basic amino acids between residues 30 and 36 of MA, a position homologous to that of MLV MA (*Murray et al., 2005*). However, we found that the number and composition of basic residues in this motif varies between ERV-Fc lineages.

Retroviral NC proteins typically encode one or more C-C-H-C motif-containing zinc finger (ZF) domains, which mediate important interactions with nucleic acids (*Chance et al., 1992*). The NC domains of betaretroviruses and lentiviruses encode two ZFs, while most gammaretroviral NCs encode a single ZF. The related HERV-H and HERV-Fc elements were previously reported as an exception among gamma-like retroviruses in encoding two ZFs in NC (*Jern et al., 2005*). Indeed, we found that the majority of the reconstructed ERV-Fc NC proteins contained two ZFs, except the ERV-Fc2 lineages present in the genomes of Hominidae and Cercopithecinae species, which have three ZF motifs in NC (*Figure 3A*).

Another important feature found in the Gag proteins of retroviruses is the late domain, which is crucial for the late stages of the retroviral replication cycle, including budding and viral release (*Göttlinger et al., 1991*; *Wills et al., 1994*; *Yasuda and Hunter, 1998*; *Yuan et al., 1999*). Late domains can be encoded by one or more of the following motifs: PPPY, P(T/S)AP, or YPXnL. Respectively, these motifs interact with the following components of the cellular endosomal sorting machinery: Nedd4, TSG101, and ALIX (*Demirov and Freed, 2004*). MLV Gag sequences contain a single copy of all three motifs within the C-terminus of MA and N-terminus of p12; however, the specific composition and location of these motifs vary considerably between extant retroviruses (*Demirov and Freed, 2004*; *Segura-Morales et al., 2005*). We identified late domain motifs in all of the reconstructed ERV-Fc Gag proteins. Similar to what has been observed for extant retroviruses, we found that the composition of these motifs and their position within Gag varied between ERV-Fc lineages (*Figure 3B*). All ERV-Fc p12 domains harbor a single PPPY motif, and most ERV-Fc Gag reconstructions also contain one or two P(T/S)AP domains. However, the position of the P(T/S)AP is flexible. Usually, these are found in p12, but in some instances a P(T/S)AP motif is present in the C-terminus of NC. The ERV-Fc Gag reconstruction from the tarsier genome is the only consensus sequence found to encode a YPXnL motif, where two such motifs are found in the p12 domain. Thus, all ERV-Fc *gag* genes are predicted to encode the essential functions known from studies of extant retroviruses. However, the particular strategies employed for doing so varied from virus to virus. These findings are consistent with evolution in the face of selective pressure to maintain these functions that would be exerted during exogenous viral replication and not from evolution due to mutations (substitutions and indels) that arose as part of a host genome following endogenization.

To further explore the evolutionary history of ERV-Fc structural proteins, we quantified the similarity of residues relative to the global ERV-Fc consensus for each position of Gag (*Figure 3C*). This analysis revealed a non-random pattern of amino acid diversity, consistent with evolutionary constraints imposed by the known or predicted functions of the viral proteins with respect to the retroviral replication cycle. The MA and CA domains, and to a lesser extent the NC domains, displayed a relatively low degree of amino acid diversity. MA and CA play critical structural/functional roles in the retroviral life cycle, and previous studies reported that the function of these proteins is highly sensitive to experimental mutational perturbations (*Auerbach et al., 2007*; *Leung et al., 2006*; *Rhee and Hunter, 1991*; *Rihn et al., 2013*; *Soneoka et al., 1997*; *von Schwedler et al., 2003*; *Yuan et al., 1993*).

In contrast, the ERV-Fc p12 region showed very low primary sequence conservation. This may not be surprising as studies of extant gammaretroviruses have shown that p12 does not appear to provide an important structural function and exhibits flexibility with regard to late domain position and interactions with host proteins (*Elis et al., 2012*; *Wight et al., 2012*).

A dichotomous pattern of amino acid conservation was observed for the ERV-Fc NC proteins. The N- and C-termini were found to be poorly conserved, while the central portion was well conserved. It is this central, conserved portion, where the essential ZF domains are located.

Structures of the N-terminal domains (NTDs) of CA have been solved for a number of retroviruses, including MLV (*Mortuza et al., 2004*). We used the Phyre2 modeling suite to predict a structural model of the global consensus ERV-Fc NTD monomer (*Kelley et al., 2015*). Six of these monomers were then overlaid onto the MLV NTD hexamer structure using PyMol (*Mortuza et al., 2004*; *Schrödinger, 2010*). Onto this structural model, we mapped the amino acid diversity at each residue of ERV-Fc (*Figures 3D and 3E*). This revealed that residues with the greatest conservation are present in regions that are known, from studies of modern retroviruses, to be involved in intramonomeric (*Figure 3D*) and intrahexameric (*Figure 3E*) contacts (*Mortuza et al., 2004*; *Gitti et al., 1996*; *Kingston et al., 2000*). Specifically, intramonomeric contacts tend to occur between faces of alphahelices oriented to the center of the CA monomer, and these are well conserved in ERV-Fc (*Figure 3D*). Similarly, intermonomeric contacts in CA hexamers are predominantly made by residues in the betahairpin and alphahelices 1, 2, and 3, which were also well conserved between the ERV-Fc consensus sequences (*Figure 3D and E*). In contrast, solvent-exposed amino acids (outside the betahairpin) tend to be less well-conserved. This may reflect either relaxed structural constraint or the fact that the CA surface can be targeted by a variety of cellular antiviral factors, such as TRIM5$\alpha$ (*McCarthy et al., 2013*; *Ohkura et al., 2011*), TRIM-Cyp (*Sayah et al., 2004*; *Brennan et al., 2008*; *Liao et al., 2007*; *Newman et al., 2008*; *Virgen et al., 2008*; *Wilson et al., 2008*), Fv1 (*DesGroseillers and Jolicoeur, 1983*; *Kozak and Chakraborti, 1996*), and MxB (*Fricke et al., 2014*; *Goujon et al., 2013*; *Kane et al., 2013*; *Liu et al., 2013*). Thus, this structural analysis shows that the pattern of amino acid diversity in ERV-Fc CA sequences is consistent with experimental mutational analyses of both MLV and lentiviral CA proteins (*Auerbach et al., 2007*; *Rihn et al., 2013*; *McCarthy et al., 2013*). As such, these analyses provide evidence that replicative pressures largely determined the observed pattern of amino acid diversity in the reconstructed ERV-Fc CA NTDs.

## ERV-Fc pol gene and the viral enzyme sequences

Similar to other gammaretroviruses and gamma-like retroviruses, the ERV-Fc *pol* gene encodes all the viral enzymatic functions, including the viral protease (PR), reverse transcriptase (RT), ribonuclease H (RNase H), and integrase (IN). All these functions are critical for successful viral replication, and as such one would anticipate a high degree of sequence conservation in *pol*. Indeed, we observed very low diversity scores across the majority of *pol*. Amino acid residues with high diversity scores were primarily found near protease cleavage sites and away from catalytic active sites (*Figure 3—figure supplement 1*).

## ERV-Fc envelope glycoproteins

The viral *env* gene encodes the proteins responsible for binding to the host entry receptor and mediating fusion of the viral and host membranes (*Hunter, 1997*). Retroviral Env proteins are expressed as a single large polyprotein which is proteolytically cleaved into the surface (SU) and transmembrane (TM) subunits by cellular furin-like proteases. The resulting complex is a heterotrimer composed of three SU subunits and three TM subunits. SU is involved in receptor binding, while TM both anchors Env in the membrane and mediates host/viral membrane fusion. As even closely related viruses often utilize different receptors for entry, and because SU is also the major target of antibodies during the host immune response, sequence diversity in SU is usually very high (*Katz and Skalka, 1990*). The extracellular portion of retroviral TM proteins creates a series of alpha helices whose orientation shift dramatically during membrane fusion (*Hunter, 1997*). Gamma-like retroviral Envs also possess a conserved disulfide motif (CX$_n$CC) involved in covalent interactions with SU, and a highly conserved immunosuppressive domain (*Bénit et al., 2001*). Due to these functional constraints, the extracellular region of TM is rather well conserved (*Katz and Skalka, 1990*; *Bénit et al., 2001*). The diversity within ERV-Fc Env sequences follows a pattern as would be expected based on prior knowledge of Env function; the SU domain displays extremely high sequence diversity, while the N-terminus of TM (where the alpha helices and the CXnCC and ISD motifs are located) is well conserved (*Figure 3—figure supplement 2*).

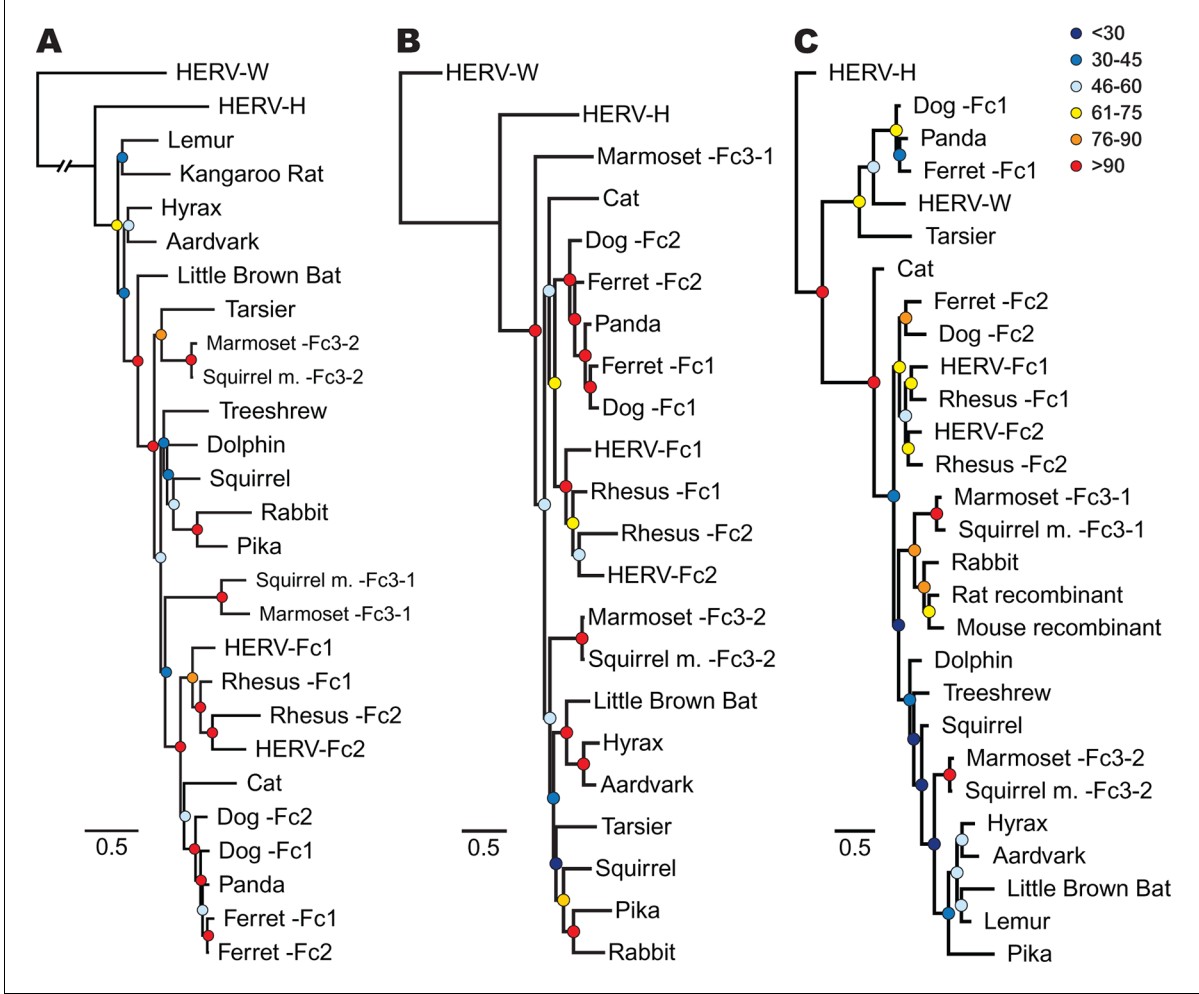

**Figure 4.** Phylogenetic relationship between ERV-Fc sequences. Maximum likelihood amino acid trees of (**A**) Gag (**B**) Pol and (**C**) TM generated using the LG substitution matrix. In each panel, HERV-H and HERV-W sequences were included as outgroups. Boostrap confidence values of nodes are depicted by colored spheres. In order to save space, a distance of approximately 0.6 was removed from the HERV-W outgroup branch in the Gag phylogeny (**A**), as indicated by the broken line.

The following source data and figure supplement are available for figure 4:

**Source data 1.** Full-length ERV-Fc Gag alignment.
**Source data 2.** ERV-Fc CA alignment.
**Source data 3.** Full-length ERV-Fc Pol alignment.
**Source data 4.** ERV-Fc RT alignment.
**Source data 5.** Full-length ERV-Fc Env sequences, including all recovered open reading frames.
**Source data 6.** ERV-Fc TM alignment.
**Source data 7.** Alignment of ERV-Fc Pol including both inferred and strict consensus sequences.
**Figure supplement 1.** Inferences made in deriving ERV-Fc consensus sequences do not significantly affect phylogenetic relationships.

## Exogenous spread of ERV-Fc involved frequent interspecies jumps and recombination

In order to track the spread and evolution of the virus, we performed phylogenetic analyses using the consensus reconstructions of all three viral precursor proteins from each species. Depending on the viral history, assessing the relationships for all viral proteins can provide either increased confidence in associations between viruses from different species or reveal lineages that have a history of recombination. Initially, we examined the evolutionary history of ERV-Fc Gag. To do so, maximum likelihood (ML) phylogenies were generated from viral Gag sequences stripped of p12, which was omitted due to extremely low amino acid sequence conservation. HERV-H and HERV-W sequences were also included as outgroups based on previous reports that these are the most closely related sequences to ERV-Fc (*Jern et al., 2005*). The analysis revealed that all the ERV-Fc Gag sequences formed a monophyletic branch distinct from HERV-H and HERV-W (*Figure 4A*). We identified several clades consisting of ERV-Fcs identified in closely related species, including Old World primates, New World primates, lagomorphs, and carnivores. However, the ERV-Fc Gag phylogeny did not recapitulate the known phylogeny of the host species – that is, some clades comprised ERV-Fc from distantly related mammals. Such patterns are suggestive of interspecies transmission (discussed below).

As retroviruses undergo a high degree of recombination, the evolutionary relatedness of the ERV-Fc sequences identified may differ based on which coding region is examined (*Bénit et al., 2001*; *Henzy and Johnson, 2013*). To gain a better understanding of the evolutionary relatedness of the identified ERV-Fc sequences, phylogenetic analyses of Pol were also performed (*Figure 4B*). Similar to what we observed for Gag, the consensus Pol sequences from Old World primates, New World primates, lagomorphs, and carnivores, which comprise distinct clades in the Gag analysis, also clustered together in the Pol phylogeny. In contrast, there were also some noticeable differences in relationships between the Gag and Pol sequences. For instance, the relationship between the carnivore ERV-Fc1 and ERV-Fc2 lineages differed in these phylogenies. In the Gag analysis, the two distinct ERV-Fc lineages (ERV-Fc1 and -Fc2) represented in the ferret genome form a monophyletic clade, and similarly the ERV-Fc1 and -Fc2 lineages from the dog genome are most closely related to each other. In contrast, the Pol phylogeny supports a closer evolutionary relationship between ERV-Fc1 lineages from the dog and ferret genomes and separately the dog and ferret ERV-Fc2 lineages, than between the two lineages from the same species. Such incongruencies in topology are an indication that recombination events involving different viral lineages took place between these two viral regions (examined further in the discussion that follows).

Finally, we examined the TM region of Env (*Figure 4C*). The SU domain of Env was not included as it is known to be one of the most rapidly evolving protein domains of retroviruses (*Bénit et al., 2001*); indeed, we found that levels of primary sequence identity within ERV-Fc SU were too low for informative phylogenetic analysis (*Figure 4—source data 5*). Similar to both Gag and Pol analyses, in the TM analysis ERV-Fc lineages from Old World primates formed a distinct clade within the tree. However, other aspects of the TM phylogeny revealed a history of recombination events involving Env. First, and most strikingly, several TMs formed a monophyletic clade with HERV-W. These include tarsier and a distinct subclade comprised of carnivore TMs (panda as well as dog and ferret ERV-Fc1). Based on the evolutionary relatedness of the Gag and Pol sequences from these species, it is likely that two recombination events occurred that resulted in these acquisitions of ERV-W Env by ERV-Fc: one leading to the ERV-Fc sequences found in the tarsier genome, and a separate event creating the chimeric virus that spread between several carnivore species and independently endogenized each of them. Second, the mouse and rat TM sequences included here clearly share identity with ERV-Fc; however, these are found in the context of two unrelated betaretrovirus-like genomes. Third, while the Gag and Pol sequences from lagomorphs consistently branched together, the TM sequences from rabbit and pika were most similar to the rat Beta-recombinant and aardvark TM sequences, respectively. Again, such incongruencies argue strongly for a history of recombination involving ERV-Fc coding regions. In summary, the phylogenetic analyses presented here argue for a long viral history of active exogenous replication during which time numerous recombination and interspecies transmission events occurred, resulting in the disparate topologies observed between the different viral protein phylogenies.

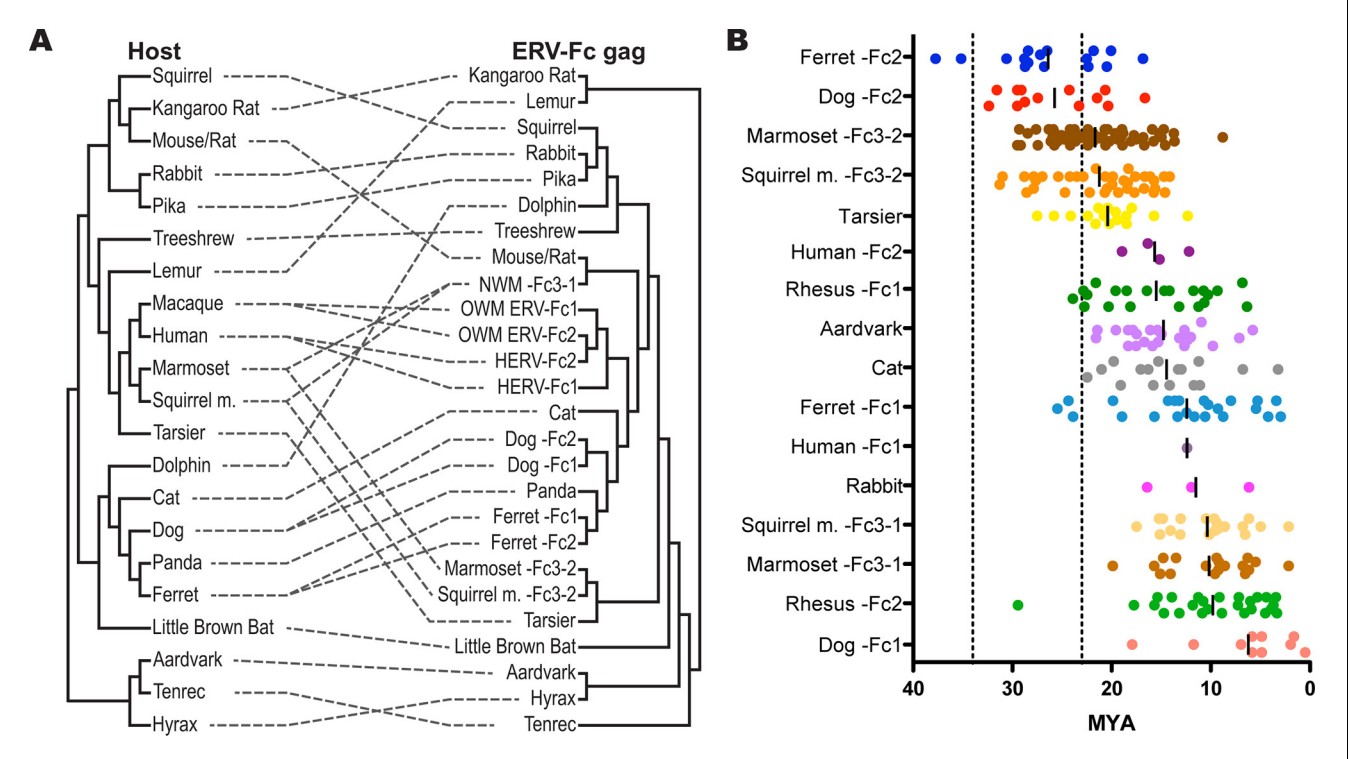

**Figure 5.** ERV-Fc has a multimillion-year history of replication with multiple cross-species transmissions. (**A**) Tanglegram comparison of host (left) and ERV-Fc phylogenies (right); dashed lines match species and the ERV-Fc found within their genome. The host phylogeny was adapted from (**Bininda-Emonds et al., 2007**), while the ERV-Fc phylogeny is a supertree generated using Matrix Representation Parsimony (MRP) based on CA and Gag amino acid phylogenies. (**B**) LTR-derived age estimates of ERV-Fc loci derived by applying a neutral evolution rate of $4.5 \times 10^{-9}$ substitutions per site per year to the nucleotide divergence between the 5' and 3' LTRs. Each plotted point represents the age estimate of a single genomic locus. Loci that show clear signatures of gene conversion or recombination have been omitted from this analysis. The average age is indicated by black vertical lines. Dotted lines indicate the approximate boundaries of the Oligocene epoch (~33.9 to ~23 MYA). ERV, endogenous retrovirus.

The following figure supplement is available for figure 5:

**Figure supplement 1.** Tanglegram comparison of host (left) and ERV-Fc phylogenies (right).

## An extensive history of cross-species transmissions involving ERV-Fc

To further examine the contribution of cross-species transmission events to the distribution of ERV-Fc among mammalian genomes, we performed a tanglegram analysis (i.e. we compared the viral phylogeny with that of the host species). The null hypothesis is that the virus co-speciated with the host (either as an exogenous virus or as preexisting endogenous elements), which would produce host and viral phylogenies with similar topologies. Deviations from the null hypothesis (co-speciation) are revealed when lines connecting each virus taxon with that of its host taxon (the genome in which it was found) cross one another, indicating instances where cross-species transmission events are likely to have occurred. The results of the comparison between the host phylogeny and ERV-Fc Gag (which allowed for inclusion of the greatest number of taxa) are shown in **Figure 5A**. For this analysis, a supertree was created using ML and Bayesian trees based on both CA or Gag (stripped of p12, as described above). This approach provided a method for inclusion of ERV-Fc isolates from tenrec, rat, and mouse, for which only a CA sequence could be retrieved (**Figure 2—source data 2**), without sacrificing analytical robustness gained by including more residues in the analysis. This analysis revealed numerous incongruencies between the phylogenies of ERV-Fc Gag and their hosts. A similar web of crossing lines was observed when the ERV-Fc Pol phylogeny was used (**Figure 5—figure supplement 1**). Furthermore, the vertical distance traversed by the connecting line can provide a proxy for estimating the relative evolutionary relatedness of the species involved in individual

cross-species transmission events. *Figure 5A* provides evidence for a number of cross-species transmission events between species of the same mammalian order (e.g. human and rhesus). Such events might be expected to predominate for several reasons – for example, there are likely to be fewer genetic barriers to viral replication in the new host (and such barriers should be easier to overcome) and closely related hosts may be more likely to be sympatric and thus more likely to encounter one another and exchange viruses (*Holmes and Drummond, 2007*; *Mollentze et al., 2014*; *Sharp and Hahn, 2011*). However, we also found evidence for cross-species transmission involving hosts belonging to different mammalian orders (e.g. tenrec, lemur, dolphin), which is thought to be much rarer due to the genetic distances involved (*Denner, 2007*). Additionally, several mammalian lineages also harbored two lineages of ERV-Fc, including Great Apes, Old World monkeys, New World monkeys, ferrets, and dogs (*Figure 5A* and Table S2). In each instance, the molecular data indicate that the two lineages originated from independent cross-species transmissions and genome colonization events in these species. These findings indicate that the distribution of ERV-Fc in the mammalian species included in this study predominantly originated following cross-species transmission events of exogenously replicating viruses.

These observations in conjunction with other molecular characteristics of the viral lineages led us to estimate a minimum of 26 independent cross-species transmission events resulted in the observed distribution of ERV-Fc among the mammalian species examined. However, this likely underestimates the total contribution of interspecies transmission to the spread of the exogenous virus because transmission may be more common between closely related species (e.g. due to genetic similarity of the hosts, or sharing of the same or similar range or niche). Such jumps between closely related hosts are less likely to be detected using incongruence between virus and host trees and this would be especially true when viral jumps occur close to speciation events.

## The history of ERV-Fc viral transmission and endogenization spans 30 million years of mammalian evolution

Estimating the time of endogenization can provide a minimal estimate of the age of the corresponding exogenous virus. To produce estimates of when active ERV-Fc endogenization occurred in the mammalian genomes examined, we employed two independent methods. First, where ERV-Fc loci originated in a common ancestor of multiple species this history of vertical transmission was used to assign upper and lower age estimates for these proviral loci. Using established time estimates of speciation (*Hedges et al., 2006*), we could place an upper bound based on the divergence time between species where one harbors a given proviral locus that a second species lacks. The lower bound was similarly determined, but in this case using divergence times between species that share orthologous loci. Second, we used a molecular clock calculation based on the divergence of the 5' and 3' LTRs of individual ERV-Fc loci (*Dangel et al., 1995*; *Johnson and Coffin, 1999*; *Martins and Villesen, 2011*) (*Figure 5B*). Due to the mechanism underlying viral reverse transcription, the two LTRs of a provirus have identical sequences at the time of insertion, and afterward these sequences acquire mutations in accordance with the neutral evolution rate of the host genome. Therefore, older proviruses would possess LTRs that are more divergent than young loci.

Our data revealed that ERV-Fc sequences in several mammalian clades reflected endogenization in a common ancestor with subsequent vertical inheritance by multiple descendant species where they were sampled in this study. Mammalian clades for which this is the case include the Great Apes (humans, chimpanzee, gorilla, orangutan); Old World monkeys (grivet, baboon, rhesus, cynomologous); New World monkeys (marmoset, squirrel monkey); and muridae (mouse, rat). Based on these findings and the reported times of speciation for these clades (*Hedges et al., 2006*), we inferred that ERV-Fc was undergoing genome colonization between 30 and 10 million years ago (MYA). For example, the presence of similar or shared ERV-Fc2 elements among Great Apes must reflect infections that occurred prior to divergence from a common ancestor 15.8 MYA (orangutan/human split), but more recent than 19.9 MYA (human/gibbon split). Shared ERV-Fc1 elements among Great Apes indicate that this viral lineage endogenized apes between 15.8 MYA (orangutan/human split) and 8.9 MYA (gorilla/human split). Similarly, the New World primate ERV-Fc3-2 found in the genomes of the squirrel monkey and marmoset must be at least 19.1 million years old (marmoset/squirrel monkey split), but younger than 43.1 million years old (Platyrrhini/Catarrhini split). Also, both the ERV-Fc1 and ERV-Fc2 lineages in Old World primates were found in grivet, baboon, and rhesus, indicating an age of at least 11.7 million years, but additional data indicated that these lineages are also

present in colobine monkeys [Patel, Senter, Johnson & Diehl, personal communication], pushing this this date back to 17.1 MYA, but no later than 29.1 MYA (Hominoidae/Cercopithecidae split). Finally, the ERV-Fcs present in mice and rat genomes pre-date speciation (22.6 MYA) but are not found in hamsters (43 MYA).

We also performed molecular clock analysis of ERV-Fc loci based on LTR divergence so that we could estimate the age of endogenization in a larger proportion of species examined in this study. For these analyses, we used a neutral evolution rate of $4.5 \times 10^{-9}$ substitutions per site per year (*Waterston et al., 2002*). This evolutionary rate is approximately twice that previously estimated for the primate/hominid lineage. However, employing the lower evolutionary rate produced age estimates suggesting insertion of ERV-Fc loci should pre-date major lineage splits where we have examined genome sequencing data and failed to find evidence corroborating such ancient insertional dates (e.g. Cercopithecidae/Hominoidae and Feliformia/Canidae/Arctoidea). Regardless, the data in *Figure 5B* illustrate two phenomena: 1) genome colonization by ERV-Fc in the species examined occurred at many times over the course of many millions of years and 2) following colonization, expansion within many lineages (by reinfection and/or retrotransposition) often continued for many millions of years.

Regarding the first point, the oldest ERV-Fc loci are the ERV-Fc2 elements in the genomes of ferret and canine, which date to 35.2 and 32.4 MYA, respectively. Nearly as old are ERV-Fc3-2 isolates from the squirrel monkey and marmoset genomes, whose oldest loci date to 31.3 and 29.3 MYA, respectively. These species diverged from a common ancestor approximately 19 MYA, and they share many orthologous ERV-Fc3-2 loci. Importantly, the LTR-based molecular clock calculations on these ERV-Fc3-2 loci yield age estimates consistent with age estimates based on species divergence times. Several species harbor ERV-Fc isolates whose oldest loci date to around 20 MYA. This group includes rabbit, human (HERV-Fc2), squirrel monkey (ERV-Fc3-1), marmoset (ERV-Fc3-1), and dog (ERV-Fc1). The ERV-Fc2 from rhesus is likely to fall in this group as well, in spite of the fact that there is a single outlier locus with a calculated age of approximately 30 MYA. We believe that this significantly overestimates the true age of this particular locus, possibly due to the relaxed evolutionary constraints of its position in the centromeric region of the Y chromosome. Nucleotide substitution rates observed on the Y chromosome and near centromeres are significantly higher than for other regions of the genome (*Brown and O'Neill, 2014*; *Hughes et al., 2010*; *Malik, 2009*; *Xue et al., 2009*). Thus, our data showed that significant cross-species transmission and endogenization by ERV-Fc took place over a span of more than 10 million years.

In addition to the long evolutionary period during which ERV-Fc was actively invading mammalian genomes, there is evidence in these data for long periods of post-endogenization amplification of ERV-Fc elements in the majority of host germ lines. For example, for nearly every species' genome that contained multiple ERV-Fc integrations, the difference in age between the oldest locus and the youngest locus was greater than 10 million years. In addition, the younger loci displayed features known to correlate with expansion by post-insertion amplification mechanisms, including large deletions of the viral genes (*Magiorkinis et al., 2012*), while the older loci tended to retain recognizable *gag*, *pol*, and *env* sequences [68 and data not shown].

## Complex history of ERV-Fc in carnivores: multiple recombination events and interspecies transmissions

The data presented up to this point suggest an interesting history for ERV-Fc within carnivores. The data shown in *Figure 4* supports the existence of two distinct lineages (ERV-Fc2 and ERV-Fc1). ERV-Fc2 sequences encode a canonical ERV-Fc Env (*Figure 4C*), are found in the cat, dog, and ferret genomes, and appeared in the genomes of the ancestors of dogs and ferrets very early in the history of ERV-Fc (*Figure 5B*). In contrast, ERV-Fc1 elements encode an Env most similar to ERV-W (*Figure 4C*), are present in the genomes of giant pandas, dogs, and ferrets, and colonized the ancestors of dogs and ferrets >10 million years after ERV-Fc2 (*Figure 5B*). Furthermore, we observed an incongruency in the dog and ferret ERV-Fc1/ERV-Fc2 relationship between the Gag and Pol phylogenies (*Figure 4*, panels A and B).

Incongruency between the Gag and Pol trees could represent either divergent selective pressures acting on these genes, or a history of recombination following the origin of these distinct viral lineages. To distinguish between these possibilities, we generated phylogenies based on nucleotide alignments of *gag* sequences from individual ERV-Fc loci from the dog and ferret genomes, along

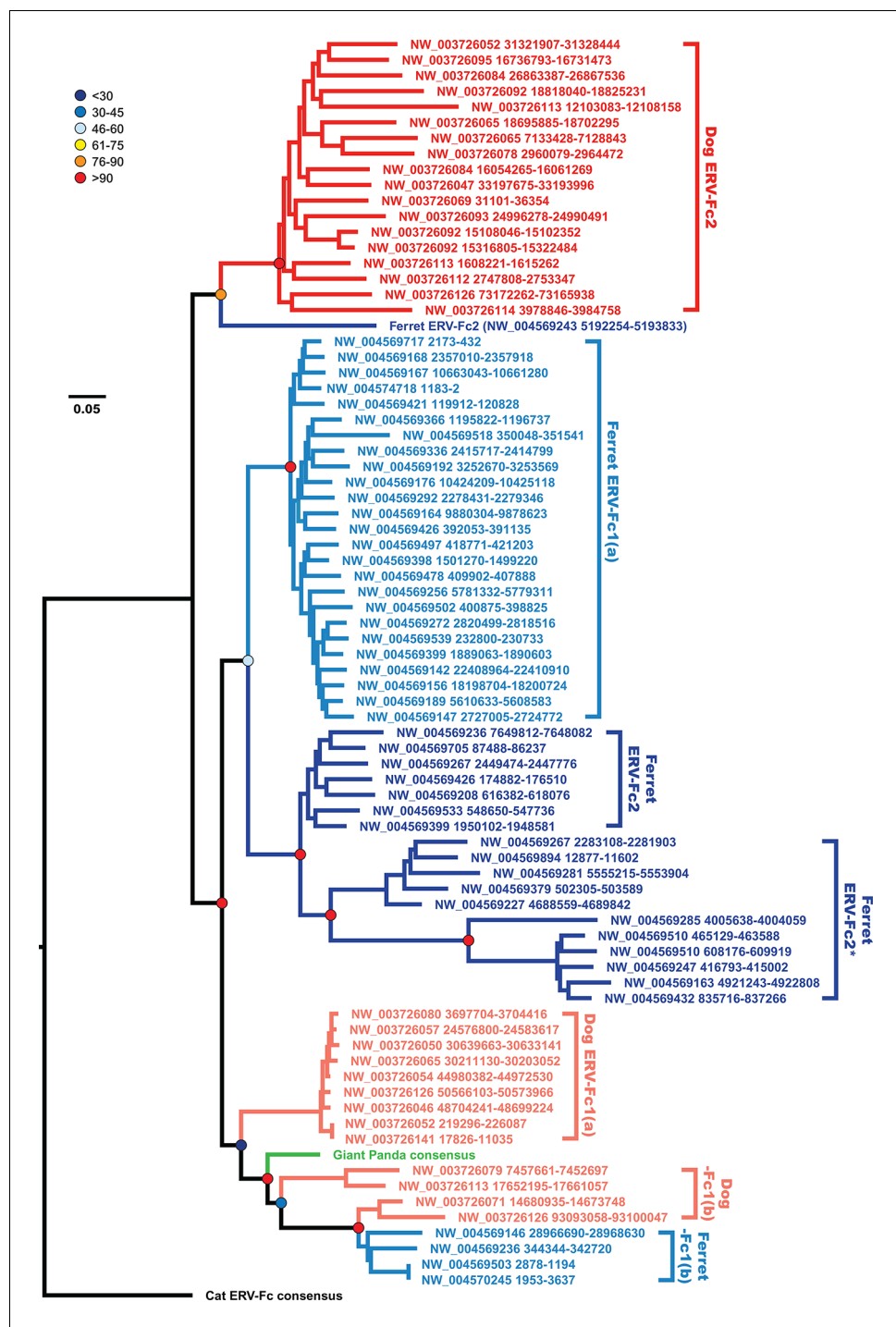

**Figure 6.** The evolutionary history of carnivore ERV-Fc1 includes numerous cross-species transmission events and at least one recombination event. Maximum likelihood phylogenetic analysis of carnivore ERV-Fc *gag* nucleotide sequences: sequences from the dog genome are colored in a shade of red, those from the ferret genome are colored in a shade of blue, and the panda consensus *gag* is colored in green. The feline ERV-Fc consensus *gag* sequence has been included as an outgroup and is colored black. For sequences from the dog and ferret genomes, the darker colored taxa are ERV-Fc2 sequences (defined based on their association with an ERV-Fc envelope sequence), while the lighter colored taxa are ERV-Fc1 sequences (defined by an association with ERV-W envelope sequence). Lineages where a large portion of the *gag* sequence has been replaced with heterologous non-coding sequence is denoted by * in the name. Boostrap confidence values of ancestral nodes are depicted by colored spheres. ERV, endogenous retrovirus.

*Figure 6 continued on next page*

*Figure 6 continued*

The following source data is available for figure 6:

**Source data 1.** Nucleotide alignment of carnivore ERV-Fc gag sequences.

with consensus giant panda and feline ERV-Fc nucleotide sequences (*Figure 6*). This analysis provided several insights into Carnivora ERV-Fc evolution. First, there is a clear ERV-Fc1 clade comprised of loci from all species known to harbor this recombinant ERV-Fc lineage (dog, ferret, and giant panda), and all canine ERV-Fc1 sequences reside in this clade. However, there are three distinct ERV-Fc1 sublineages present within the dog genome that have distinct relationships with ERV-Fc1 sequences from other carnivores. We interpret this as an indication of at least two, but more probably three, separate cross-species transmission events into dogs. Also found in this ERV-Fc1 clade is a minor population from the ferret genome [ERV-Fc1(b)]. However, the majority of the ferret ERV-Fc1 *gag* sequences form a monophyletic clade with *gag* sequences of the non-chimeric ferret ERV-Fc2. The observed phylogenetic associations are evidence that a recombination event replaced *gag* within the ferret lineage. This is supported by the observation that all ERV-Fc1 *pol* sequences, including those from ferret, form a monophyletic clade (data not shown). Finally, a solitary ferret ERV-Fc2 *gag* forms a monophyletic clade with the canine ERV-Fc2 sequences. This likely indicates that at least two distinct lineages of ERV-Fc2 jumped from another species into an ancestor of the ferret lineage: one potentially originating in a canine ancestor, and a second coming from an unknown source.

Combined, our data support a natural history of ERV-Fc1 such as that depicted in *Figure 7*. Briefly, an initial recombination event allowed for the acquisition of an ERV-W envelope by a carnivore ERV-Fc-related virus. Subsequently, multiple cross-species transmission events resulted in colonization of the genomes of the ancestors of dogs, ferrets, and giant pandas. Up to three interspecies transmission events account for the genetic diversity of ERV-Fc1 presently found in the canine genome. Due to the relatively close evolutionary relationship between ERV-Fc1(b) loci found in the ferret genome and one subset of ERV-Fc1(b) in the canine genome, it is possible that this lineage was transmitted directly between the ancestors of dogs and ferrets. However, a similar relationship could also be explained by independent transmissions of similar viruses from a third, unidentified species. Following introduction of ERV-Fc1(b) into mustelids, it appears that additional recombination event(s) took place with a pre-existing ERV-Fc2 virus, or viruses, resulting in the acquisition of the ERV-Fc2 *gag*. It was this double chimeric ERV-Fc1 that most successfully invaded the ferret genome.

## Discussion

ERV loci can be used to reconstruct the natural history of the ancient, exogenously replicating retroviruses. Previous studies examining retroviral macroevolution via the ERV fossil record have cast an wide net, focusing primarily on the highly conserved RT as a phylogenetic marker and using it to characterize a broad swath of diversity within the Retroviridae family (*Jern et al., 2005*; *Hayward et al., 2013*; *2015*). However, focusing on RT excludes additional sources of phylogenetic signal available to resolve relationships between closely related taxa, and may overlook the potential role that recombination plays in retroviral evolution. Thus, we sought to examine the deep evolutionary history of a single retrovirus lineage – that which produced the ERV-Fc family of sequences – by collecting and analyzing endogenous retroviral sequence information for all three of the canonical retroviral genes (*gag, pol*, and *env*). Doing so allowed us to identify ERV-Fc sequences in 28 of the 50 mammalian genomes examined. Furthermore, we determined that as many as 26 independent cross-species transmission events produced the distribution of identified ERV-Fc elements. This included several species whose genomes appeared to have been independently colonized by two evolutionarily distinct ERV-Fc lineages. These results indicated that the distribution of ERV-Fc among modern mammals is predominately the result of interspecies spread and emergence of the related exogenous forms of the virus.

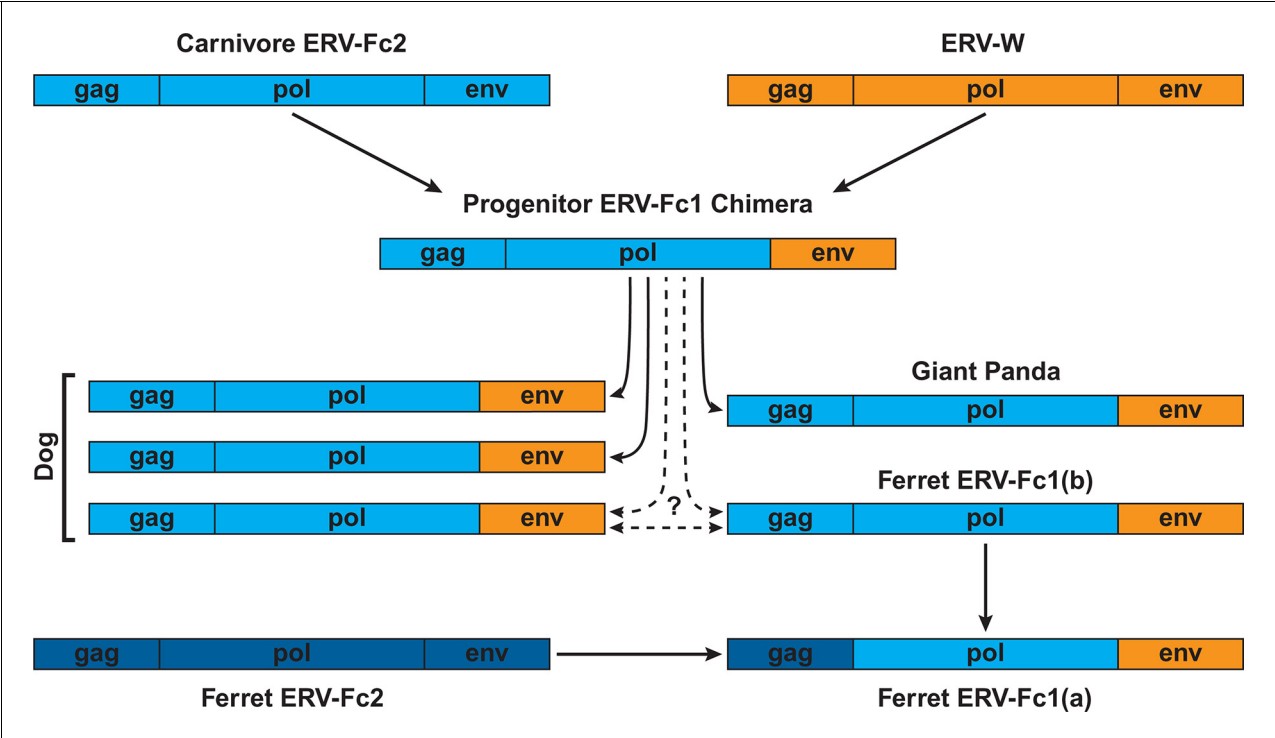

**Figure 7.** Proposed recombination and transmission sequence involving carnivore ERV-Fc1. ERV-Fc sequences are depicted in blue, while ERV-W sequences are depicted in orange. See text for a detailed explanation of the arrows.

ERV sequences present in the genomes of different species can be related either due to vertical inheritance (as genomic loci) or due to independent colonization by an exogenous, infectious virus. The two scenarios differ primarily due to differences in the rate at which exogenously replicating virus sequences and endogenous sequences evolve, as well as any differences in the selective pressures affecting parasitic genomic elements versus those affecting replicating viruses. We found that the patterns of amino acid diversification between ERV-Fc sequences were consistent with selection to maintain functions essential for exogenous viral replication. For example, the critical structural subunits of Gag (MA and CA) displayed the least diversity, and within CA, the most conserved residues were in regions involved in essential intrahexameric interactions. In contrast, primary sequence conservation in the non-structural subunits p12 and NC was significantly lower. In spite of this diversity, these regions retained their critical, canonical motifs, instead the number and location of these motifs varied significantly between viral isolates. This is consistent with selection to maintain essential motifs in a system that otherwise lacks structural constraint.

The abundance of ERV-Fc sequence information allowed us to explore the evolutionary relationship between, and infer the history of, the individual ERV-Fc lineages uncovered. Our analyses point to a complex life history for the ERV-Fc retroviral lineage. This history began >30 MYA and exogenous replication continued for many millions of years and involved multiple cross-species transmission events. Recent studies have found evidence for cross-species transmissions in examinations of endogenous gammaretroviruses that are similar to extant MLV, and some of the jumps that these viruses made appear to have involved distantly related host-species (*Hayward et al., 2013*; *2015*). Taken together, gamma-like retroviruses appear to have had a rich history of cross-species transmissions that contrasts to the life histories of other retroviral genera. For example, exogenous foamy viruses are known to co-speciate with their hosts and the endogenous record suggests that long-term associations between foamy viruses and their hosts are likely to be an ancient feature of this retroviral genus (*Han and Worobey, 2012*; *Katzourakis et al., 2009*; *Switzer et al., 2005*).

Furthermore, our analyses revealed that recombination played an important role in the life history of ERV-Fc with instances of acquisition of *pol* and *env* sequences from HERV-H or HERV-W-like

viruses, as well as evidence that an ERV-Fc-related virus provided *env* sequence to a betaretrovirus. In this regard, the recombination observed within the carnivore ERV-Fc1 clade of viruses is noteworthy. In this lineage, an ERV-W *env* gene replaced the ancestral ERV-Fc *env*; subsequently, this chimeric virus was involved in at least two, and possibly as many as five, cross-species transmission events, giving rise to the endogenous sequences found in the genomes of modern dogs, ferrets, and giant pandas. The Pol and TM regions of the chimeric virus, ERV-Fc1, form monophyletic clades, clearly indicating a shared ancestry for the viruses in the different species. However, within the dog and separately the ferret genome, the Gag sequences ERV-Fc1 and ERV-Fc2 are more closely related to one another than they are to sequences of the same lineage from the heterologous species. The recombinant ERV-Fc1 lineages were also observed to be younger than the majority of ERV-Fc2 loci in both dog and ferret genomes. Thus, the data revealed a scenario whereby after cross-species transmission the ERV-Fc/ERV-W *env* chimera acquired the ERV-Fc2 *gag* present in the genome of its new host species, in this case an ancestor of modern ferrets. Such a scenario would be consistent with the virus acquiring the ability to either interact with positive acting host proteins or avoid host restriction factors, or both.

Our analysis suggests that the origins of ERV-Fc date back at least as far as the beginning of the Oligocene epoch (~33.9 MYA). This was a time period of dramatic global change marked by the fusion of the African to the European as well as the Indian to the Asian continental plates (*Briggs, 1995*), climatic cooling, development of vast expanses of grasslands, and the emergence of large mammals as the world's predominate fauna (*Prothero and Berggren, 1992*). Continental mergers in the Old World along with a continued Asian-North American connection allowed for significant mammalian migrations throughout the Oligocene. However, we found evidence for ERV-Fc being present in species with little or no known geographic overlap at this early time in the viral life history, including musteloids, canids, Platyrrhini, and Tarsioidae. This makes it difficult to pinpoint a geographic region for the origin of the ERV-Fc viral lineage, as the ancestors of modern species whose genomes harbor ERV-Fc were geographically isolated from each other at the time. The fossil record provides solid evidence that during the Oligocene epoch canids were restricted to North America (*Munthe, 2005*), musteloids were present in Asia (*Sato et al., 2012*), and Platyrrhini were likely restricted to South America (*Bond et al., 2015*). The previously widespread distribution of Tarsioidae, which were found in Africa, North America, Europe, and Asia, was contracting to its current geographic isolation in southeast Asia (*Gingerich, 2012*). The geographic separation of these host species, coupled with the clear phylogenetic relationships between their viral sequences, provides strong evidence for a rapid global spread of the exogenous forms of ERV-Fc. Based on current phylogeographic knowledge of these early ERV-Fc hosts, and evidence for limited faunal exchange between these continents, we find it unlikely that musteloids, canids, Platyrrhini, or prosimians were solely responsible for this global viral spread. Importantly, the ERV-Fc genomic record in modern mammalian genomes likely represents only a fraction of the total exogenous viral spread: for example, exogenous infections may simply have failed to leave an endogenous footprint in some species, and some unknown proportion of lineages bearing ERV-Fc insertions will have eventually become extinct (and the corresponding ERV-Fc record lost). Thus, it is likely that the ERV-Fc "fossil" record is incomplete, and that either extinct species or species lacking ERV-Fc sequences helped facilitate the worldwide spread of the exogenous virus.

Finally, our results indicate that after the birth of ERV-Fc, replication, cross-species transmission, and endogenization continued for approximately another 15 million years. Our data, as well as other published reports (*Bénit et al., 2003*; *Barrio et al., 2011*), indicate that active ERV-Fc reinfection may have continued until very recently in some lineages, indicating that at least one ERV locus has retained functional *gag* and *pol* coding potential in those species. In ferret and canine, we found evidence that ERV-Fc1 acquired *gag* sequence from an older, pre-existing endogenous lineage (ERV-Fc2). Indeed, LTR dating indicated that in both species the oldest ERV-Fc1 locus pre-dates the end of active reinfection of the genome by ERV-Fc2. Thus, it is plausible that there existed in the genomes of the ancestors of these species at least one functional ERV-Fc2 *gag* ORF that the newly introduced ERV-Fc1 could have acquired. Observations in laboratory mice as well as in vitro and in vivo experiments provide several well characterized examples of recombination involving ERV sequences giving rise to replication-competent viruses with novel biological properties (*Chong et al., 1998*; *Coffin et al., 1989*; *Paprotka et al., 2011*; *Patience et al., 1998*; *Telesnitsky and Goff, 1993*). Thus, ERV loci could contribute to adaptive evolution of exogenous

viruses by providing a reservoir of novel sequences that can be tapped into by co-packaging and recombination.

## Materials and methods

### Identification and extraction of ERV-Fc loci from mammalian genomes

Previously published human and baboon ERV-Fc sequences (*Bénit et al., 2003*) were used as bait in Basic Local Alignment Search Tool (BLAST) queries of 50 mammalian genome-sequencing databases hosted at the National Center for Biotechnology Information (NCBI) (Table S1). These genomes comprise a broad sampling of metatherian and eutherian mammals including representatives of every continent except Antarctica. However, as indicated in Table S1, these genomes were generated using various approaches that result in ERV sequence information of varying quality. High-coverage Sanger sequencing approaches employing BAC and whole-genome shotgun sequencing utilized to produce genomic sequence for the human and mouse, amongst others, results in maximally reliable assemblage of non-coding sequences. In contrast, genomes assembled using low-coverage Sanger sequencing or only Illumina sequencing data often possess fragmentary non-coding sequence information due to difficulties in accurately assembling short stretches of sequence covering repetitive sequences.

To extract maximal sequence information from genomic databases, an iterative BLAST approach was used. Initial BLAST queries utilized CA, RT, and/or TM inputs and were used to perform nt BLAST searches with the following parameters: match/mismatch scores of +1/−1; gap costs of 0 and 2 (existence, extension); and repeat masking turned off. BLAST hits were extracted along with surrounding sequence information, and these were analyzed using RepeatMasker (Institute for Systems Biology, Seattle, WA [http://www.repeatmasker.org]) to identify sequences of ERV origins, and those of ERV identity were in turn used as query sequences with the same scoring parameters as the initial BLAST query. Then, BLAST hits from individual genes were correlated in an attempt to identify complete, or mostly intact, genomes. Approaches for extracting ERV sequences from the genomic databases diverged at this point. In the few instances where the genome sequencing data allowed, ERV LTR sequences were identified, and these were used to extract genomic regions flanked by two LTRs, the vast majority of which represented ERV genomes. Alternatively, if ERV genomes could be identified, they were used as query sequences for a subsequent BLAST interrogation with match/mismatch scores of +2/−3 and gap costs of 5 and 2 (existence, extension). If full-length ERV sequences could not be identified, due to either the specific genomic sequence being of too low quality or the lack of intact ERVs in the genome, full-length *gag, pol*, and *env* sequences were used in the final round of BLAST interrogation with the same parameters as used with full-length ERV sequences. When full-length *gag, pol*, or *env* sequences could not be obtained, then sequences extracted from the second round of BLAST interrogation (using initial CA, RT, or TM hits as query sequences) were utilized.

### Generation of ERV-Fc consensus sequences

Extracted ERV sequences were aligned, initially using the MUSCLE algorithm (*Edgar, 2004*) as implemented in Geneious 6 (Biomatters, Auckland, NZ) and then further refined by hand. Consensus sequences were generated from these alignments; however, when multiple disparate types of ERV sequence were retrieved, separate consensus sequences were generated for each class. The consensus sequences were then used to infer the proteins encoded by the ERVs. This was done by replacing premature termination codons with the most frequently represented alternate codon and substituting 'R' or 'Y' ambiguities (transition mutations) to either cytosine or guanine at locations determining the amino acid encoded (i.e. NOT third base wobble positions). If consensus sequences contained transversion ambiguities, multiple amino acid consensus sequences were generated, differing only at those positions.

### Generating HERV-H and HERV-W consensus sequences

To generate outgroups for phylogenetic analyses, HERV-H and HERV-W sequences were extracted from the human genome. To do this, Repbase (*Jurka et al., 2005*) sequences corresponding to *gag, pol*, and *env* of HERV-H and HERV-W (listed as HERVH and HERV17 in Repbase, respectively) were

used as BLAST queries to interrogate the human genome with match/mismatch scores of +2/−3 and gap costs of 5 and 2 (existence, extension). BLAST hits were extracted and aligned, initially using the MUSCLE algorithm as implemented in Geneious 6 with further refining by hand. Consensus generation proceeded as for ERV-Fc.

## Phylogenetic analyses

Multiple alignments were generated from the consensus ERV-Fc, HERV-H, and HERV-W protein sequences. These included CA, RT, and TM subunit alignments for all identified ERV sequences and separately full-length Gag, Pol, and Env alignments for the subset of sequences that possessed full-length sequence information. Sequences can be found in *Figure 4—source datas 1–6*. CA alignments comprised approximately 225 amino acids spanning from the N-terminal proteolytic cleavage site to a conserved poly-charged region upstream of the CA/MA proteolytic cleavage site. For phylogenetic analysis, the Gag alignment was trimmed of the p12 region, as sequences in this region possessed low primary sequence homology and varied greatly in length. RT alignments consisted of approximately 216 amino acids that spanned from the N-terminal QΦP that forms portion of the DNA binding domain to 10 amino acids C-terminal of the conserved ΦLG involved in the catalytic function. TM alignments included approximately 130 residues of extracellular sequence from the poly-charged furin cleavage site to a conserved tryptophan adjacent to the poly-hydrophobic putative transmembrane sequence.

Phylogenetic trees were constructed using both Bayesian Markov chain Monte Carlo (MCMC) and ML algorithms as implemented in MrBayes 3.2.1 (*Huelsenbeck and Ronquist, 2001*) and PhyML (*Guindon and Gascuel, 2003*), respectively. ML phylogenies were generated for each alignment using combined NNI/SPR (nearest neighbor interchange/subtree pruning and regrafting) searching optimizing for topology, branch length, and substitution rate parameters, with the proportion of invariable sites set to 0.0, four substitution rate categories, and an estimated gamma distribution parameter. Support for the ML branching patterns was assessed by performing 200 bootstrap replicates. Separate ML trees were generated for each alignment using the LG (*Le and Gascuel, 2008*) and RtREV (*Dimmic et al., 2002*) amino acid substitution models. Bayesian phylogenies were calculated using the Poisson rate matrix, gamma rate variation with four gamma categories, with unconstrained branch lengths. Two parallel MCMC analyses of 1,100,000 steps each were performed using four heated chains and a heated chain temperature of 0.2. Sampling of the trees was performed every 200 trees and omitting the first 500 trees (100,000 steps). Effective sample sizes of more than 600 indicated convergence of the MCMC run. In many instances, we included multiple ERV-Fc sequences for individual species. One source of this was due to uncertainties in determining an ancestral ORF sequence, which we resolved by generating multiple consensus sequences each reflecting different possibilities at ambiguous residues (as described above). The other reason for multiple consensus sequences derives from the fact that some species harbored multiple distinct lineages of ERV-Fc, and in these cases, we generated independent consensus sequences for each lineage. Phylogenies were generated including these multiple consensuses and were subsequently collapsed for publication when all formed a monophyletic branch.

To assess the influence our method of reconstructing ancestral ORFs had on the generated phylogenetic topologies, 'strict' consensus sequences were generated for all ERV-Fc lineages where we were able to reconstruct a full-length Pol. When necessary, the 'strict' consensus sequence was edited in order to maintain frame. Otherwise, all ambiguities and premature stop codons were maintained. Alignments were produced as per above, and phylogenies were generated via RAxML v7.2.8 (*Stamatakis, 2006*) as implemented in Geneious 8 (Biomatters, Auckland, NZ) using the LG substitution model and the GAMMA model of rate heterogeneity. Confidence analysis was performed via 100 bootstrap replicates. This set of analyses were performed using RAxML instead of PHYML because inclusion of premature stop residues are forbidden in PHYML.

Supertrees were generated using Clann (ver. 3.2.3) (*Creevey et al., 2004*; *Creevey and McInerney, 2005*) with six total input trees: three based on CA sequence and three based on Gag (- p12) sequence. These CA and Gag inputs include two ML phylogenies, one generated using the RtREV substitution model and the second using LG, and a single Bayesian phylogeny. To reflect the fact that Gag phylogenies are based on a larger dataset of parsimonious information, the input trees were weighted where Gag=2 and CA=1. Heuristic searches were performed using Matrix Representation Parsimony (MRP) with subtrees generated using either the SPR (sub-tree pruning and re-

grafting) or TBR (tree bisection and reconnection) resampling. Minimal differences were observed in phylogenies produced using these resampling methodologies, and phylogenies produced by TBR are presented here.

Gag nucleotide alignments were produced of ERV-Fc1 and ERV-Fc2 sequences from carnivores. This included sequences from individual proviral loci from the ferret and dog genomes as well as the ERV-Fc consensus sequences generated from the cat and giant panda genomes. Similar to the amino acid alignment described above, the p12 region was stripped from the alignment as it showed signs of genetic saturation. ML phylogenies were produced in MEGA5.2 using the GTR (general time reversible) substitution model, pairwise deletion of gapped data ('use all sites' option), and NNI topology optimization.

## Constructing host/virus tanglegrams

The Gag supertree and Pol ML phylogeny were further utilized as viral input trees for constructing virus-host tanglegrams. The host phylogeny used in generating these tanglegrams was generated by pruning a previously published pan-mammalian phylogeny (*Bininda-Emonds et al., 2007*). TreeMap 3 was employed to deconvolute the relationship between the viral and host phylogenies and produce the fewest number of incongruencies (as indicated by crossed lines) (*Charleston and Robertson, 2002*).

## Calculating ERV-Fc diversity and mapping it to protein structure

Gag, Pol, and Env ERV-Fc protein sequences were assessed for amino acid composition. As millennia of exogenous and endogenous replication were likely to have allowed these viruses to explore a nearly infinite evolutionary space, a means for examining the diversity of amino acid properties at a given residue was utilized. This system utilizes the Smith and Smith penalty matrix where identical residues are given a 0 penalty, highly similar residues (such as Ile and Leu) are given a penalty of 1, residues sharing some chemical properties (such as Leu and Phe) are given a penalty of 2, and completely dissimilar residues (such as Leu and Pro) are given a penalty of 3 (*Smith and Smith, 1990*). Diversity scores at each residue were calculated by summing pairwise Smith and Smith scores between a global ERV-Fc consensus and all ERV-Fc sequences.

Global ERV-Fc consensus protein sequences were used to model structures of individual protein domains. Models were generated using the **P**rotein **H**omology/analogy **R**ecognition **E**ngine V 2.0 (Phyre$^2$) server, which uses PSI-Blast, hidden Markov modeling, and profile-profile matching algorithms to thread novel amino sequences onto known protein structures (*Kelley and Sternberg, 2009*). For multimeric structures, the Phyre ERV-Fc model was overlayed onto MLV structures. In the case of assembling the CA hexamer, the 1U7K (*Mortuza et al., 2004*) MLV structure was utilized as a scaffold. Individual residues were then colored according to their calculated diversity scores.

# Acknowledgements

We would like to thank Evan Senter and Adam Jenkins for creating computer scripts to automate data retrieval and analysis and Kevin McCarthy for assistance with structural predictions. Work in the Johnson laboratory is supported by Boston College and by NIH grants AI083118 and AI095092.

# Additional information

## Funding

| Funder | Grant reference number | Author |
| --- | --- | --- |
| National Institutes of Health | AI083118 | Welkin E Johnson |

The funders had no role in study design, data collection and interpretation, or the decision to submit the work for publication.

## Author contributions

WED, Conception and design, Acquisition of data, Analysis and interpretation of data, Drafting or revising the article; NP, Acquisition of data, Analysis and interpretation of data, Drafting or revising

the article; KH, Conception and design, Acquisition of data, Drafting or revising the article; WEJ, Conception and design, Analysis and interpretation of data, Drafting or revising the article

**Author ORCIDs**

Welkin E Johnson, http://orcid.org/0000-0001-5991-5414

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
