## [Decision Letter]

Thank you for submitting your work entitled "Tracking interspecies transmission and long-term evolution of an ancient retrovirus using the genomes of modern mammals" for consideration by *eLife*. Your article has been favorably evaluated by Detlef Weigel (Senior editor) and three reviewers, one of whom, Steve Goff, is a member of our Board of Reviewing Editors.

The reviewers have discussed the reviews with one another and the Reviewing Editor has drafted this decision to help you prepare a revised submission.

Summary:

This paper reports the history of a particular family of gammaretroviruses, deduced from mining the fossil record of endogenous retroviral DNAs in the genomes of a large set of modern mammals. The data reveal the interspecies spread of this virus family, recombination events occurring during its spread, and evolution of viral genes plausibly in response to pressures of cross-species adaptation.

Our reviewers were all positive about the story and support going forward with the paper. Some were very excited indeed! We include their full comments below. There were only a few issues raised that require attention, which we would like to see addressed in a revised version.

Essential revisions:

A highly knowledgeable reviewer (#3) questions if the conclusions are fully supported and raises a key issue about the analysis: s/he requests further bootstrap analysis of subsets of the sequences (as I understand it – see the review), to show how robust the trees and conclusions are. One would want to know that the same basic story falls out, no matter what subset is used in the analysis. Along these lines, we need to know if the same general conclusion follows with analysis of a few specific genomes, instead of the consensus.

*Reviewer #1:*

This paper reports the history of a particular family of gammaretroviruses, deduced from mining the fossil record of endogenous retroviral DNAs in the genomes of a large set of modern mammals. The data reveal the interspecies spread of this virus family, recombination events occurring during its spread, and evolution of viral genes plausibly in response to pressures of cross-species adaptation.

The search of genome sequences revealed ERV-Fc members in 28 species, and their phylogeny based on Gag or Pol was not consistent with a single introduction followed by divergence as an ERV but rather requires multiple independent transmission events (more than 26!). Two major lineages were apparent, each with complex histories.

Strengths:

The paper is notable for the wide range of host species examined and the final conclusion that this virus has been surprisingly active in invading genomes so many times. Figure 5 is the money shot. I think this is a significant story.

The paper is also notable in including trees based not only on Pol but also Gag and Env sequences, and so revealing recombination.

Weaknesses:

There is a previous report along these lines (Benit, Calteau and Heidmann, 2003). But it offered a very limited analysis.

The Discussion drags somewhat. I think the paper could be tightened up a lot.

*Reviewer #2:*

This paper describes an in-depth investigation into one group of mammalian endogenous retroviruses (ERVs), referred to as 'ERV-Fc'. The work reported here consists of: (i) the recovery of the ERV-Fc 'fossil record' (comprised of the various ERV-Fc-derived sequences scattered throughout mammalian genomes); (ii) use of fossil sequences to reconstruct a canonical ERV-Fc genome, complete with annotations based on comparison to contemporary retroviruses; (iii) an investigation of the evolutionary history of ERV-Fc in mammals. In my opinion the methods used are appropriate and the manuscript is distinguished by a high standard of presentation and methodological rigor throughout.

Advances in DNA sequencing have enabled the emergence of paleovirology – the study of ancient viruses. With remarkable speed, we have progressed from a situation in which the challenge was to obtain any data about ancient viruses, into one where data is accumulating faster than it can be analyzed. This is a happy circumstance for researchers in most ways, but unfortunately does encourage a flag-planting approach to publication, in which authors rush to lay claim to discoveries but avoid the more painstaking and difficult work of interpreting them. I was therefore greatly impressed by the meticulousness and depth of this investigation, which through painstaking and detailed analysis constructs a detailed picture of an ancient retrovirus species.

ERV-Fc sequences scattered widely and heterogeneously throughout the mammalian germ line are used to define the period of geologic time in which this ancient retrovirus spread, thus providing insight into the kind of ecological circumstance in which it existed. Phylogenetic analysis reveals an intriguing pattern of interspecies transmission and a complex history of recombination. These detailed investigations provide the platform for discussion of certain fundamental concepts surrounding the processes that generate ERV diversity, and the interpretation of ERV fossil data. Although the significance of the many surprising aspects of ERV-Fc biology revealed here (such as the complex recombination history in carnivores) is not yet known, this paper establishes firm foundations for future investigations, including experimental studies.

In terms of methodological rigor and overall clarity, I think this is a field-defining paper. It will be immensely valuable for the way that it communicates certain key concepts about interpretation of the ERV fossil record that have not to my knowledge been discussed elsewhere, that are not necessarily intuitive, and have often been misunderstood.

Finally, despite being a highly accomplished piece of work, the paper retains a cautious and modest tone throughout, and does not attempt to overstate its conclusions, which also impressed me.

*Reviewer #3:*

The authors present a comparative study tracing the descent of an endogenous retrovirus lineage (ERV-Fc) through mammalian species over an estimated 30 million years. Regular and iterative BLAST searches were used to identify ERV-Fc sequences. Consensus sequences were inferred for Gag, Pol, Env, as possible, and phylogenetic and related evolutionary analysis was performed on consensus alignments. The authors' analysis suggests rapid viral spread to diverse species involving frequent cross-species transmission and recombination events, in addition to vertical transmission following endogenization. While cross-species transmission and recombination among ERVs is known, the study is interesting for its comprehensive focus on a specific ERV lineage and the manuscript is generally well presented. Unfortunately it is not clear from major parts of the analysis how well the conclusions are supported.

It isn't clear the quality of the consensus sequences generated and how much they were altered to "undo" mutations and create amino acid alignments used for phylogenetic analysis. Perhaps the bigger issue is a lack of bootstrap analysis, or indication of posterior probabilities for Bayesian trees, for the phylogenetic analysis using maximum likelihood, which puts the inferences of cross-species transmission into question and many of the proposed recombination events into question if the phylogenetic arrangements are not robust. Similar concerns apply to the 'supertree' combining arrangements of CA or Gag sequences. Related to this, alignments should be made available as a supplementary dataset.

Additionally, was similar, independent analysis considered with at least a subset of recovered ERV-Fc sequences versus consensus sequences?

---

## [Author Response]

*A highly knowledgeable reviewer (#3) questions if the conclusions are fully supported and raises a key issue about the analysis: s/he requests further bootstrap analysis of subsets of the sequences (as I understand it – see the review), to show how robust the trees and conclusions are. One would want to know that the same basic story falls out, no matter what subset is used in the analysis. Along these lines, we need to know if the same general conclusion follows with analysis of a few specific genomes, instead of the consensus.*

We were very pleased with the positive comments of all three reviewers, and we are grateful for their constructive critiques. We have modified the manuscript in order to incorporate their suggestions, which amounted to small modifications to the figures, textual edits (to improve readability, as requested) and inclusion of some additional supplemental material. The suggestions made by reviewer #3 were particularly helpful in strengthening the reliability of our overall conclusions, and we begin by providing our responses to his/her comments:

Reviewer #3:

*The authors present a comparative study tracing the descent of an endogenous retrovirus lineage (ERV-Fc) through mammalian species over an estimated 30 million years. Regular and iterative BLAST searches were used to identify ERV-Fc sequences. Consensus sequences were inferred for Gag, Pol, Env, as possible, and phylogenetic and related evolutionary analysis was performed on consensus alignments. The authors' analysis suggests rapid viral spread to diverse species involving frequent cross-species transmission and recombination events, in addition to vertical transmission following endogenization. While cross-species transmission and recombination among ERVs is known, the study is interesting for its comprehensive focus on a specific ERV lineage and the manuscript is generally well presented. Unfortunately it is not clear from major parts of the analysis how well the conclusions are supported. It isn't clear the quality of the consensus sequences generated and how much they were altered to "undo" mutations and create amino acid alignments used for phylogenetic analysis.*

This was a key concern from the beginning of our study – while reconstructing accurate reading frames was essential, we also recognized that it was important to do so without introducing unintentional biasinto the sequences. The consensus sequences used in our analyses (referred to here as “inferred” consensus) were very conservatively changed from that of the “strict” consensus sequences. In order to illustrate how nearly identical the inferred and strict consensus sequences are, we produced a full-length Pol derived phylogeny that includes “strict” consensus sequences for all ERV-Fc lineages (Figure 4—figure supplement 1; the source alignment is included as [Supplementary-material SD10-data]). In all cases, the matching “inferred” and “strict” consensus sequences cluster, with only very small length differences appearing in the tip branches, which represent those minor differences that were introduced intentionally when we “undid” mutations to make the inferred consensus reconstructions.

*Perhaps the bigger issue is a lack of bootstrap analysis, or indication of posterior probabilities for Bayesian trees, for the phylogenetic analysis using maximum likelihood, which puts the inferences of cross-species transmission into question and many of the proposed recombination events into question if the phylogenetic arrangements are not robust.*

We apologize for the oversight. We now include the bootstrap support values for the nodes on all phylogenetic trees.

*Similar concerns apply to the 'supertree' combining arrangements of CA or Gag sequences.*

The reviewer raised a very important point, prompting a careful reevaluation of the supertree and associated analyses. While we felt that a single supertree was a convenient way to present all the underlying analyses, reexamination suggests that this is not as robust as using the original individual CA and Gag source trees. The method (Average consensus, or AvCon) uses pairwise evolutionary distances between taxa to create phylogenies that are blind to the phylogenetic relationships present in the source trees, and the resulting phylogenies did have differences from the CA and Gag source trees – specifically, the generated supertrees had altered placement of ERV-Fc sequences from the tarsier and little brown bat genomes. Due to this, the supertree in Figure 4 has been replaced with an ML phylogeny based on ERV-Fc Gag.

As the tanglegram presented in Figure 5 was based on the same AvCon phylogeny as Figure 4, this was also replaced in the revision. Here, we utilized a supertree generated using matrix representation parsimony. In this case, the resulting supertree faithfully represents the topology of the source trees and was suitable for the tanglegram analysis shown in the figure (this tree was not used in Figure 4 because it lacks branch length information).

Related to this, alignments should be made available as a supplementary dataset.

We now include the alignments for Gag, CA, Pol, RT, and TM as supplemental data files. Additionally, we have uploaded un-aligned, full-length Env sequences, including both consensus Env sequences as well as all intact Env open reading frames (ORFs).

*Additionally, was similar, independent analysis considered with at least a subset of recovered ERV-Fc sequences versus consensus sequences?*

To address the possibility that our “inferred consensus” sequences might impart bias, we did two things: first, we compared our results to trees built with strict consensus sequences (see response to the first comment above), and second, we did as the reviewer suggests for our detailed examination of the relationship between individual recovered sequences from carnivores (see Figure 6), which gave the same result as using inferred consensus sequences.

In addition, in response Reviewer #1, the text of the Results and Discussion has been edited to be more concise.